# Temporal Transcriptome Analysis Reveals Dynamic Expression Profiles of Gametes and Embryonic Development in Japanese Flounder (*Paralichthys olivaceus*)

**DOI:** 10.3390/genes12101561

**Published:** 2021-09-30

**Authors:** Xiancai Hao, Qian Wang, Jilun Hou, Kaiqiang Liu, Bo Feng, Changwei Shao

**Affiliations:** 1Key Lab of Sustainable Development of Marine Fisheries, Ministry of Agriculture and Rural Affairs, Yellow Sea Fisheries Research Institute, Chinese Academy of Fishery Sciences, Qingdao 266072, China; best_hxc@163.com (X.H.); wangqian2014@ysfri.ac.cn (Q.W.); liukq@ysfri.ac.cn (K.L.); fengbo_1220@foxmail.com (B.F.); 2Laboratory for Marine Fisheries Science and Food Production Processes, Qingdao National Laboratory for Marine Science and Technology, Qingdao 266071, China; 3Beidaihe Central Experiment Station, Chinese Academy of Fishery Sciences, Qinhuangdao 066100, China; jilunhou@126.com

**Keywords:** transcriptome analysis, embryo development, maternal gene, zygotic gene, Japanese flounder

## Abstract

The maternal-to-zygotic transition (MZT) is a crucial event in embryo development. While the features of the MZT across species are shared, the stage of this transition is different among species. We characterized MZT in a flatfish species, Japanese flounder (*Paralichthys olivaceus*). In this study, we analyzed the 551.57 GB transcriptome data of two types of gametes (sperms and eggs) and 10 embryo developmental stages in Japanese flounder. We identified 2512 maternal factor-related genes and found that most of those maternal factor-related genes expression decreased at the low blastula (LB) stage and remained silent in the subsequent embryonic development period. Meanwhile, we verified that the zygotic genome transcription might occur at the 128-cell stage and large-scale transcription began at the LB stage, which indicates the LB stage is the major wave zygotic genome activation (ZGA) occurs. In addition, we indicated that the Wnt signaling pathway, playing a diverse role in embryonic development, was involved in the ZGA and the axis formation. The results reported the list of the maternal genes in Japanese flounder and defined the stage of MZT, contributing to the understanding of the details of MZT during Japanese flounder embryonic development.

## 1. Introduction

The occurrence of vertebrate animals begins with the zygote by the fertilization of egg and sperm, and in this process, the egg and sperm each provide a set of chromosomes [1]. In addition, the egg provides important maternal genes, such as mitochondrial genes, polymerase and messenger RNA (mRNA). Then, the fertilized egg produces larvae that are completely different from fertilized egg in structure and function.

After fertilization, the egg does not immediately start the transcription of the zygotic genome until the completion of one to more than a dozen mitosis [2]. The transformation of gene expression patterns of a fertilized egg switches from maternal genes to zygotic genome called “maternal-to-zygotic transition (MZT)”, which encompasses two overlapping processes, including the elimination of maternal gene products and zygotic genome activation (ZGA) [3]. There is a complex interaction in the regulation of embryo development, which may overlap at the time of MZT occurrence [2,4,5]. The stage of maternal gene product elimination and the ZGA are not conserved among different species. In mice, the maternal factor degradation occurred at the 2-cell stage of embryonic development, and the minor wave of ZGA was detected in 1-cell embryos. A major wave of ZGA occurs in the 2-cell embryos [6,7,8]. In *Drosophila*, maternal gene products were degraded immediately after egg activation. The minor wave ZGA occurred in the 8-cell division of early embryos, and large-scale transcription occurred at 14-cell division [9,10]. Fast-developing embryos in vertebrate, such as amphibians and fish, often exhibit late ZGA stage [2]. In *Xenopus*, activation of zygote genomes can be detected when cells are divided into 128-cell stages. When the embryo is divided into the 13th division cycle (blastula stage), the major wave of ZGA occurs, and the maternal gene products are degraded significantly [3]. Other research reports have observed the minor wave of zygotic transcription between the 64-cell and 256-cell stages in zebrafish (*Danio rerio*) [11,12,13]. The stage of elimination of maternal gene products and large-scale zygotic activation occurs in the 10th division cycle (blastula stage) in zebrafish [14,15].

The major developmental transition of vertebrate embryos is the formation of the anterior-posterior and dorsal axes [16,17]. Wnt signaling pathway is highly conserved in different species. It plays crucial roles in the ZGA and the axis formation during embryonic development [18], inhibiting canonical Wnt signaling in *Xenopus* and causing mis-regulation of genes activated at ZGA [19]. Wnt/*β*-catenin signaling pathways are involved in the dorsal axis specialization. In zebrafish and *Xenopus*, *β*-catenin has a distinct function before and after MZT. In pre-MZT, *β*-catenin accumulates in the embryo dorsal part and activates the dorsal tissue gene, while the *β*-catenin moves to the ventral after MZT, which is necessary for inhibiting the dorsal tissue gene [20,21,22,23]. The temporal expression of *β*-catenin is crucial for forming the dorsoventral axis.

Previous studies revealed extensive variability stages among different species of maternal gene product degradation and zygotic gene activation. However, the stage of MZT in Japanese flounder (*Paralichthys olivaceus*) remains unknown. The Japanese flounder is one of the important flatfish species with the most extreme asymmetric body morphology of vertebrates [24]. In this study, we reported the transcriptomic profiling of the Japanese flounder gametes and embryo development. The RNA-seq analysis revealed that large-scale zygotic activation began at the LB stage. The Wnt signaling pathway plays multiple roles in the ZGA and axis formation during embryo development.

## 2. Materials and Methods

### 2.1. Ethics Statement

Japanese flounder used in this study were sampled from BEIDAIHE central experimental station, Qinhuangdao City, China. All the experiments were performed in accordance with the guidelines for the Care and Use of Laboratory Animals of the Chinese Academy of Fishery Sciences.

### 2.2. Embryo Preparation and Sample Collection

The eggs and sperms were obtained from 3-year-old Japanese flounder female and male broodstock by artificial ovulation. The sperms was collected using a 5-mL plastic syringe by gently pressing on the abdomen and stored in the dark at 4 °C for further use. The eggs were collected by gently stripping the abdomen of female fish into a 1000 mL glass beaker, stored in the dark at room temperature and then artificially fertilized to obtain fertilized eggs. The fertilized eggs were incubated in clean seawater with continuous aeration at 15 ± 1 °C. The fish embryos at different developmental stages were prepared according to the method described by Liu et al. [25]. The samples of unfertilized eggs (F), sperms (M) and 10 development time points of the embryo from 4-cell (4c, 2 h after fertilization, haf), 32-cell (32c, 4 haf), 128-cell (128c, 5 haf), high blastula (HB, 6 haf), low blastula (LB, 11 haf), early gastrula (EG, 15 haf), late gastrula (LG, 27 haf), neurula (NE, 32 haf), heart-beating (HBT, 55 haf) and hatching stage (HS, 62 haf) were collected and stored in liquid nitrogen. About 1000 embryos were collected at each developmental stage for each group. Before sampling, dead embryos were removed, and developmental stages were observed under a stereomicroscope (Olympus). RNA sequencing (RNA-seq) was performed on each eggs/sperms and embryo at different stages, with two biological and technical repetitions.

### 2.3. RNA-seq Library Preparation

The total RNA was extracted from each sample using Trizol (Invitrogen, Carlsbad, CA, USA). Then, the total RNA was qualified and quantified using a Nano Drop and Agilent 2100 bioanalyzer (Thermo Fisher Scientific, Waltham, MA, USA). A total of 46 cDNA libraries were constructed by the NED kit (Illumina, San Diego, CA, USA) according to the manufacturer’s instructions, and the high-quality RNA [3 μg RNA (RIN > 7.0) from each sample] was used for mRNA library construction by means of the conventional protocol. The library was sequenced on a BGISEQ-500 platform. The raw reads were deposited in China National GeneBank DataBase (CNGBdb) (BioProject: CNP0000304).

### 2.4. Data Filtering and Genome Mapping

We removed adapters and filtered low-quality reads (the rate of reads whose quality value ≤ 10 is more than 20%) by SOAPnuke (version 1.4.0); the reads with unknown nucleotides ≥ 5% (options = -l 10, -q 0.1, -n 0.05, -Q 2 -G 1) were used to obtain clean data [26]. Subsequently, the read quality was controlled by fastQC (http://www.bioinformatics.babraham.ac.uk/projects/fastqc (accessed on 25 September 2020). The clean data were aligned with hisat2 (version 2.1.0) [27] and mapped to the Japanese flounder genome [24].

### 2.5. Gene Expression and Transcriptome-Wide Time-Series Analyses and Identification of Differential Expression Genes (DEGs)

The expression level of the gene was estimated using salmon-SEME [28], and then the gene expression level was normalized by Transcripts Per Kilobase Million (TPM) to eliminate the influence of different length genes. The different expression genes were identified from DESeq2 across samples with 2 fold change [29]. Genes with similar expression patterns were clustered according the R package of Mfuzz (version 2.52; Options = −c 6, −m 1.25) [30].

### 2.6. Identification of the Maternal and Zygotic Genes

The maternal genes were detected from which gene was expressed (TPM > 2) in eggs and silent in sperms (TPM < 2) [31], and the zygotic gene detected that expression of zygotic gene increased above that of the unfertilized egg, with fold changes ≥ 2 and a false discovery rate-adjusted *P* (*q* value) < 0.05 using Deseq2 [2]. The zygotic genes were selected based on a previous report on *Xenopus* [32]. We used blastp to identify the homologous genes in *Xenopus,* zebrafish and Japanese flounder. The transcriptome data regarding the embryonic development of *Xenopus* and zebrafish were obtained by Meng How Tan et al. [33] and Richard J White et al. [34].

## 3. Results

### 3.1. Transcriptome Sequencing, Cluster Analysis of Gametes and Embryo Development in Japanese Flounder

We collected two types of gametes (sperm and egg) and ten embryo development stages for RNA-seq (Figure 1A). A total of 551.57GB raw bases were obtained. After filtering the raw reads, we generated a mean of 12.0 GB clean bases of each sample (Appendix A), and then clean-read mapping was done with Japanese flounder genome. Subsequently, 6993-8887 genes were expressed (TPM > 2) in sperms, eggs and ten embryo developmental stages. Based on the expression of each gene per sample, we conducted principal component analysis (PCA). These samples were divided into two clusters, and one cluster contained egg, 4c, 32c, 128c and HB stage samples. Other clusters included LG, NE, HBT and HS stage samples (Figure 1B).

### 3.2. The Performance of Gene Expression during Embryo Development

We identified the differentially expressed genes (DEGs) among gametes and embryonic development stages. The DEGs between sperm and egg were 5421, which indicates that the gene expression profiles of sperm and egg are different. In addition, the number of DEGs (7653) between sperm and the 4-cell stage was significantly higher than the number of DEGs (2076) between egg and 4-cell stages (*p* < 0.05) (Figure 2A). During embryonic development, the number of DEGs in the LB vs. EG stage was the highest (8759), and the number of DEGs at the 128-cell stage to HB stage was the lowest (47). At the 4-cell and HB stages, the number of DEGs between the two consecutive stages was relatively lower. After the HB stage, the DEGs between the two consecutive stages increased. We identified specific DEGs during two consecutive periods by Venn map. The highest number of specific DEGs occurred at M-VS-4c (1277), followed by the stage of HB to LB (736) (Figure 2B).

### 3.3. Pattern of Maternal mRNA products Degradation in Japanese Flounder Embryo

We identified 2512 genes involved in maternal mRNA degradation during embryonic development. Then, we employed Mfuzz to detect time-series expression profiling of the candidate genes in embryonic development (Figure 3A). These clusters exhibited genes with specific expression patterns and were classified into six clusters (C1–C6). Cluster 1 (239 genes) and Cluster 3 (521 genes) expression levels increased from the zygote stage, reached the peak at the EG and HB stage and then decreased sharply. The gene expression levels in Cluster 2 (690 genes) and Cluster 5 (449 genes) steadily risen after fertilization and then extremely degraded from the HB and LB stage, respectively. Cluster 4 (522 genes) expression level decreased after the 4-cell stage and showed a minimum level at the EG stage. Besides, the expression trend of Cluster 6 (91 genes) showed an increasing trend after the LG stage. Heatmap showed that most maternal genes maintained a high expression level from zygote to LB stage, followed by a decreased expression after the LB stage (Figure 3B).

The maternal genes, such as POU class 5 homeobox 1 (*pou5f1*), yes-associated protein 1 (*yap1*) and alpha-adducin (*adducin*), were highly expressed in the egg, after fertilization, high expression levels of those genes maintained from the stage of 4c to LB stage and expression level was dropped sharply after the LB stage. The nanog homeobox (*nanog*) maintained high expression levels during the 4c to HB stage, and the c-type lectin domain family 4 member E (*clec4e*) was high expression in stage of 4c, 128c and HB stage, both of *nanog* and *clec4e* was silence after the HB stage. The DNA (cytosine-5)-methyltransferase 1 (*dnmt1*) expression level increased at egg and 4-cell and 32-cell stages during the embryonic stage (Figure 3C).

### 3.4. Identification of Stage of ZGA and Its Association with Expression Change

To determine the stage of ZGA in Japanese flounder, we obtained 1231-6723 up-regulated DEGs, as shown in Figure 4A. The number of up-regulated DEGs in F-vs-LB (4127) was more than 1.5-fold higher than in F-vs-HB (2746). Then, we removed the 2512 maternal genes and calculated the average expression of the remaining 8826 genes. As shown in Figure 4B, the average expression decreased from the 4-cell stage, reached the lowest level at the 128-cell stage and then increased slowly. At the stage of LB, the expression level was slightly higher than that at the 128-cell stage. Then, the average expression increased significantly from the LB stage to the EG stage (*p* < 0.05) and reached the highest at the NE stage (Figure 4B).

We selected the 36 zygotic genes in *Xenopus* [32]. Most of the homologous genes did not express before the ZGA stage (at stage9 (blastula stage), dome stage (blastula stage)) in *Xenopus* and zebrafish, but started to express after the ZGA stage. Similarly, most of these genes did not express before the LB stage in Japanese flounder and began to express after the LB stage (Figure 4C). The DEGs in HB vs. LB stage contained zygotic genes, such as transcription factor AP-2-alpha (*tfap2a*) and grainyhead-like protein 3 (*grhl3*), having low expression in early embryonic development and high expression at the LB stage.

### 3.5. The Roles of Wnt Signaling Pathway in Embryo Development

We investigated the roles of the Wnt signaling pathway during embryonic development in Japanese flounder. We identified 211 genes of Wnt signaling pathways (pov04310) (Figure 5A). Of these, 150 genes were expressed (TPM > 2) during embryonic development. The 85 genes were highly expressed in the 4-cell to LB stages, and the lowly expressed after LB stage, such as *dvl2*, *dvl3*, *lrp6* and *wnt5a* (Figure 5B). Twenty-two genes were highly expressed after the stage of LG. Thirty-one genes were highly expressed at the stage of LB and EG, such as *wnt11*, *axin* and *frizzled7*, the *β*-catenin is the most important gene in the Wnt signaling pathway, and it was been detected expressed in 4c stage and highly expressed in LB to EG stage (Figure 5B).

## 4. Discussion

In this study, our data examined the gene expression profiles in gametes and embryonic development in Japanese flounder. We identified the maternal- and ZGA-related genes, and the maternal gene expression patterns showed the elimination after LB stage. In addition, the major wave of ZGA occurred at the LB stage. We also found that the Wnt signaling pathway might be involved in the ZGA and the axis formation during the embryonic development in Japanese flounder. This results enhance our understanding of the feature of the ZGA stage and the transcriptional diversity of the Wnt signaling pathway.

We first detected the dynamic expression changes in the transcriptome of egg, sperm and 4-cell embryo to hatching embryo in this study. The zygote at the 4-cell stage after fertilization was compared with sperm and egg, and the number of DEGs in sperm (7653) was more than three times that in egg (2076). The PCA results indicated that the gene expression pattern at the 4-cell stage was similar to the egg stage, which verifies that the gene expression patterns of the early zygotes are inherited from eggs. From 4-cell to HB stage, there were a few numbers of DEGs between the two consecutive periods. These results indicate that the maternal genes found in eggs play critical roles in the early embryonic development of Japanese flounder. From the HB stage to the LB stage, the number of DEGs increased, and the number of stage-specific DEGs was high at the embryonic development stage, which reveals that MZT may occur at the HB stage to LB stage in Japanese flounder embryo.

Large maternal gene products play crucial roles in early embryonic development [35]. However, with embryonic development, maternal gene products begin to degrade [36]. Maternal gene product clearance is one of the major molecular activities in MZT. In our study, we found 2512 maternal genes in Japanese flounder embryo. The expression levels of almost all maternal genes were high in early embryonic development and attenuation at the stage of LB (Figure 3B), indicating that the elimination of maternal gene products might occur at the LB stage. The stage of the maternal gene products clearance in Japanese flounder was similar to zebrafish, which occurred at the blastula stage [14,15]. A large number of maternal genes provide a rich resource for future functional studies of Japanese flounder transcription in early embryo development. In a previous study, the maternal genes, including *clec4e* and *adducin1,* were expressed in egg and then silent at the gastrula stage [37]. We further found that the *clec4e* and *adducin* were silent earlier than those at the gastrula stage (Figure 3C). The *clec4e* silence after the HB stage and the *adducin1* silence after the LB stage showed that the maternal gene products were eliminated earlier than gastrula in Japanese flounder embryo. We verified that after fertilization, the *pou5f1* and *nanog* genes were highly expressed until LB and HB stages, respectively and then weakly expressed. In zebrafish, the *pou5f1* and *nanog* are responsible for the clearance of maternal mRNAs, which activates zygotic gene expression during the MZT [11,38], implying that *nanog* and *pou5f1* may also be involved in the elimination of maternal gene products in Japanese flounder. In mice embryo, the *yap1* was highly expressed at the early stage. Deletion of *yap1* was delayed in development, and most of the embryos stopped developing during the morula stage [39]. In the present study, we found that *yap1* was sharply decreased from the LB stage to the EG stage (Figure 3C), which suggests that *yap1* plays an important role in early embryonic development and maternal gene product elimination. Other maternal genes (e.g., *dnmt1*) are involved in the methylation imprints. The lack of the *dnmt1* resulted in complete demethylation of all imprinted loci at blastocysts in mice [40], which were highly expressed in early Japanese flounder embryos in our study. The expression levels of *dnmt1* were decreased after the 128C stage, which indicates that the Japanese flounder may have completed methylation reprogramming at the 128-cell stage.

The large-scale transcription of zygote genes is another important event during MZT. In this study, we detected the stage of zygotic genome activation in Japanese flounder embryo. Unfertilized eggs were compared with different embryonic stages. The number of up-regulated DEGs sharply increased during the LB stage (Figure 4A). Similarly, the number of up-regulated DEGs increased at the stage of elimination of maternal gene products. The expression levels of genes increased at the 128-cell stage and increased sharply at the LB stage (Figure 4B). So, Thus, we deduced that zygotic genome activation might begin at the 128-cell stage and large-scale transcription at the LB stage. Furthermore, we focused on 36 zygotic genes defined by Chen et al. in *Xenopus* [32]. Most genes were not expressed at the early stage of embryonic development and expressed after ZGA in zebrafish and *Xenopus*. In Japanese flounder, most of those genes were expressed after the LB stage (Figure 4C), implying that ZGA might occur during the LB stage in Japanese flounder. In zebrafish, the zygotic gene of *tfap2a* maintains early neural crest induction and differentiation and contributes differentiation of all three germ layers [41]. The onset of *tfap2a* gene expression in the embryo LB stage of Japanese flounder reached the peak at the EG stage, indicating that it might play a similar role in zebrafish. Another zygotic gene, *grhl3* plays a crucial role in neural tube closure and regulates epithelial migration in zebrafish embryos. Over-expression in gastrulation may result in axial duplication and impaired neural tube migration leading to cyclopia [42]. In Japanese flounder, the *grhl3* was highly expressed during the LB stage to LG stage, which was the stage of formation of embryo axial. Finally, we observed that the large-scale transcriptome of zygotic genes occurred at the LB stage in Japanese flounder.

The Wnt pathway plays different roles before and after the MZT mediated by the intracellular transducer *β*-catenin (*ctnnb1*). The *β*-catenin expression commenced at the 128-cell and 16-cell stages in zebrafish and *Xenopus*, respectively. Both genes expression was local at the vegetal pole and participated in the early embryonic polarity. At the stage of later blastula and gastrulation, *β*-catenin was switched from the activator to the repressor of embryo dorsal-specific genes [43,44]. In Japanese flounder embryo, the *β*-catenin was detected at the 4-cell stage, which was detected earlier than zebrafish and *Xenopus*, suggesting that *β*-catenin played a unique role in flatfish. Then, the *β*-catenin expression level gradually increased after fertilization and reached the peak at the EG stage. The EG stage is a critical stage of dorsoventral axis formation in embryonic development. In *Xenopus*, inhibition of the canonical Wnt/*β*-catenin pathway caused limited activation of the zygotic gene, and the embryo did not form the antero-posterior axis in gastrulation after knockout [19]. We also found that the *wnt11* and *frizzled7* were highly expressed in the LB stage and the EG stage. Studies indicate that the antisense oligonucleotide-mediation of maternal *wnt11* and *frizzled7* plays an important role in dorsoventral axis specification [45,46]. According to our results, the Wnt pathway in Japanese flounder embryonic was also involved in the ZGA and dorsum axis formation. This diversity expression pattern indicates that a function for the Wnt pathway is essential for embryonic development.

## 5. Conclusions

We depicted a dynamic expression profile of embryo developmental stages in Japanese flounder. We verified that the zygotic genome transcription might occur at the 128-cell stage, and large-scale transcription began at the low blastula stage in Japanese flounder. The Wnt signaling pathway showed a diversity function on the ZGA and axis formation. These results provide insights into the conserved MZT among different vertebrate with the fast-embryo development and present their characteristics in different species during embryonic development. However, future studies are required to determine the role of these maternal- and ZGA-related genes on embryonic development.

## Figures and Tables

**Figure 1 genes-12-01561-f001:**
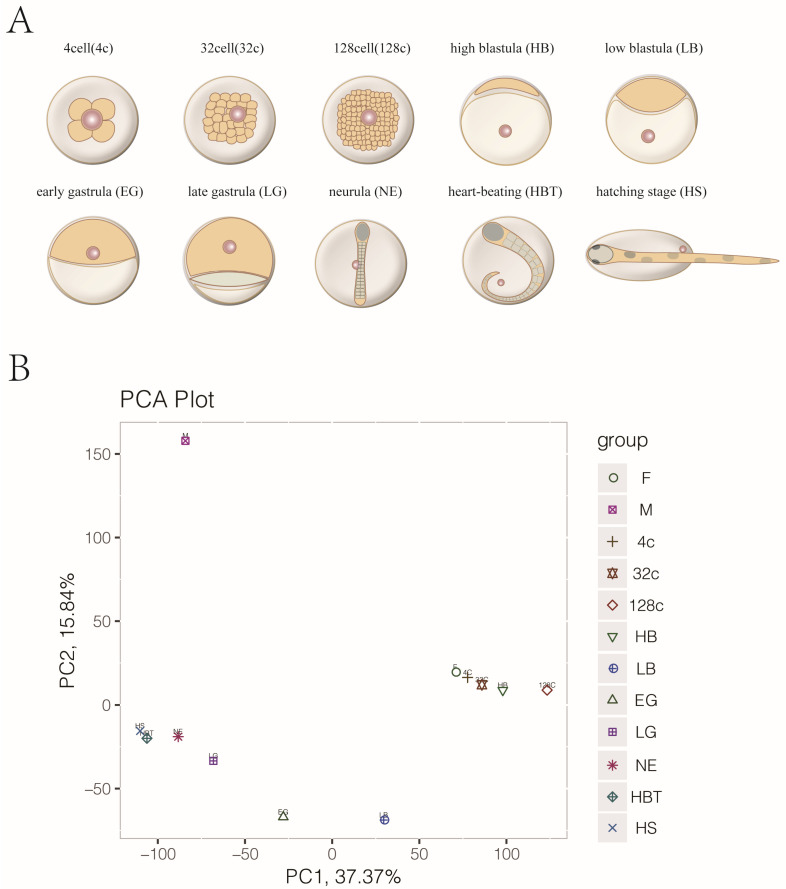
The samples collection and clustering analysis during embryonic development. (**A**): A scheme of Japanese flounder embryonic developmental stages. (**B**): Principal component analysis (PCA) showed a clear cluster during gametes and embryo development. M: sperm, F: egg, 4c: 4-cell, 32c: 32-cell, 128c: 128-cell, HB: high blastula, LB: low blastula, EG: early gastrula, LG: late gastrula, NE: neurula, HBT: heart-beating, HS: hatching stage.

**Figure 2 genes-12-01561-f002:**
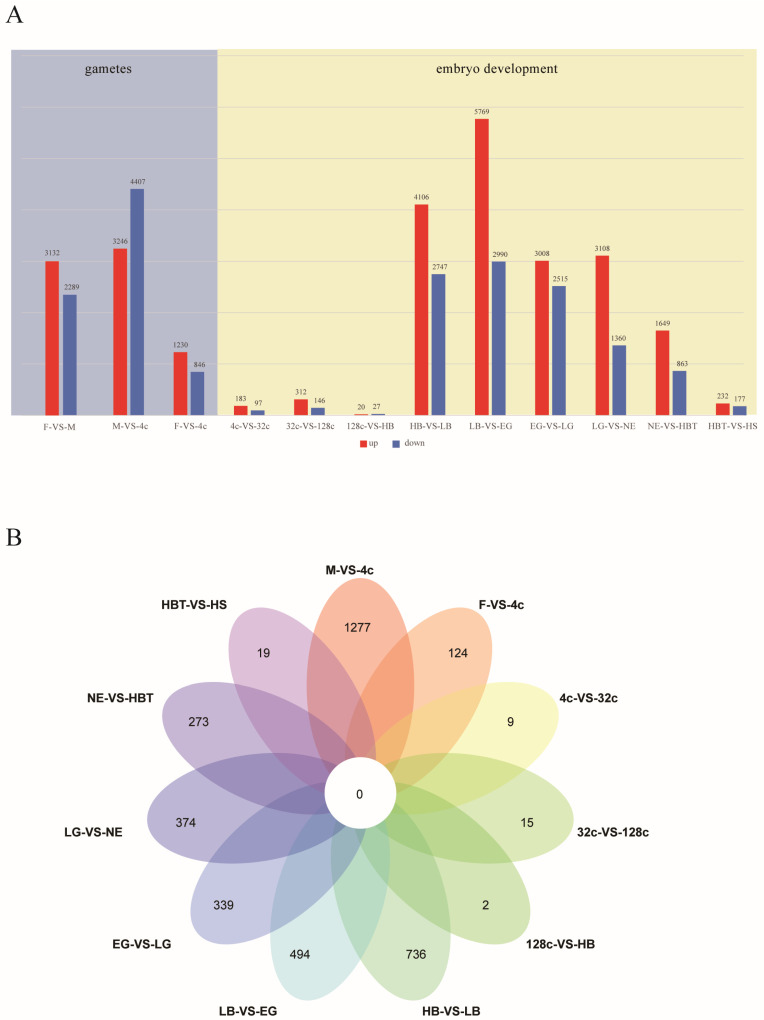
The dynamic expression profile of embryo developmental stages in Japanese flounder. (**A**) The number of differentially expressed genes (DEGs) identified between two adjacent developmental stages; Red and blue colors indicate up-regulated and down-regulated genes at later stage vs. early stage; (**B**) Venn diagram depicts the stage-specific DEGs. M: sperm, F: egg, 4c: 4-cell, 32c: 32-cell, 128c: 128-cell, HB: high blastula, LB: low blastula, EG: early gastrula, LG: late gastrula, NE: neurula, HBT: heart-beating, HS: hatching stage.

**Figure 3 genes-12-01561-f003:**
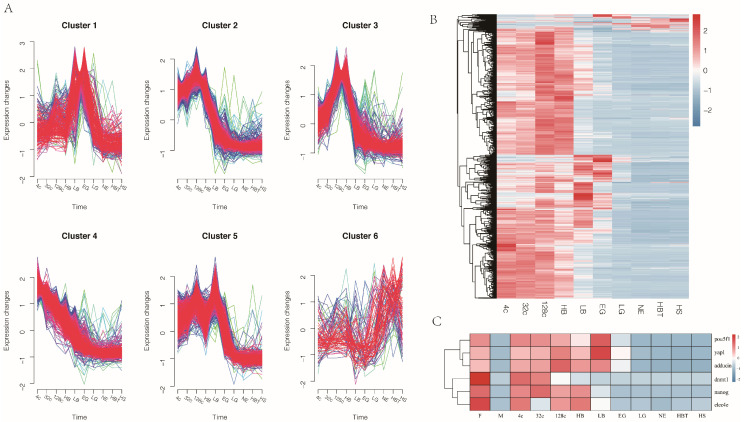
Transcriptome-wide time-series cluster of maternal gene expression profiling. (**A**) Cluster analysis of maternal gene expression based on Mufzz. (**B**) Heatmap shows the expression of the maternal genes in embryo development. (**C**) Heatmaps show expression trends of six maternal-related genes during gamete and embryonic development in RNA-seq data. M: sperm, F: egg, 4c: 4-cell, 32c: 32-cell, 128c: 128-cell, HB: high blastula, LB: low blastula, EG: early gastrula, LG: late gastrula, NE: neurula, HBT: heart-beating, HS: hatching stage.

**Figure 4 genes-12-01561-f004:**
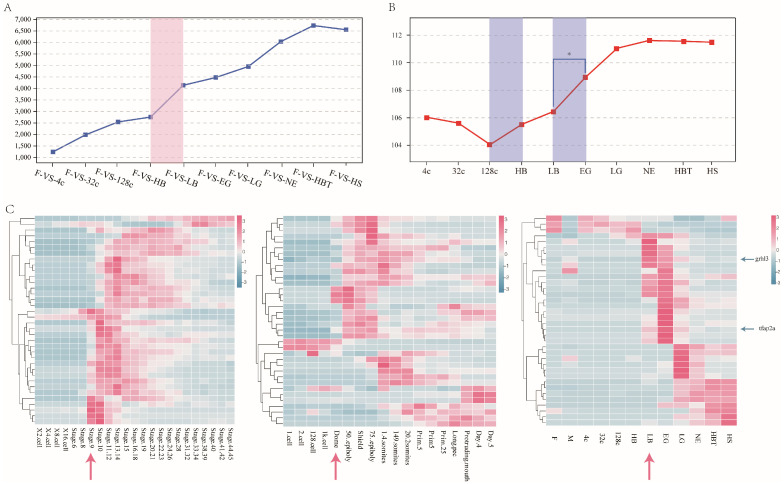
The transcription of zygotic genes during the embryonic stages. (**A**) The number of up-regulation DEGs compare zygote stage (each embryo development stage) with egg. (**B**) The average gene expression level of zygotic genes (remove maternal genes) during embryo development. * *p* < 0.05 means a statistically significant difference. (**C**) The 36 zygotic gene expression patterns during embryonic development of *Xenopus*, zebrafish and Japanese flounder. The arrow refers to the activation period of the ZGA. stage9: blastula stage, Dome: blastula stage, M: sperm, F: egg, 4c: 4-cell, 32c: 32-cell, 128c: 128-cell, HB: high blastula, LB: low blastula, EG: early gastrula, LG: late gastrula, NE: neurula, HBT: heart-beating, HS: hatching stage.

**Figure 5 genes-12-01561-f005:**
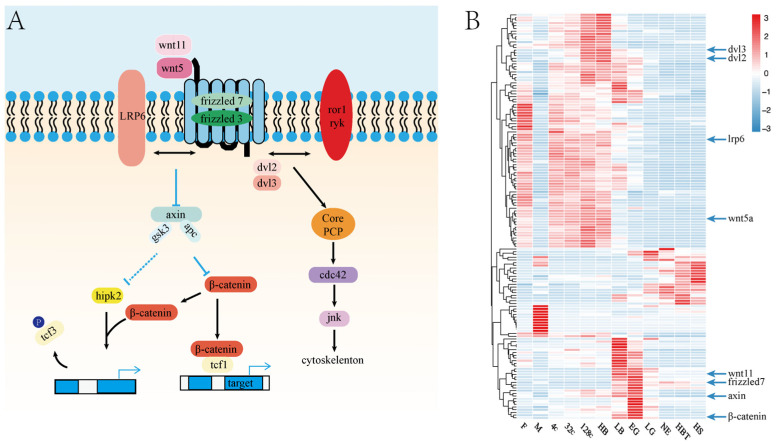
Wnt signal pathway expression patterns in embryo development. (**A**) Conserved Wnt pathway branches and components and the core PCP indicate planar cell polarity (PCP); (**B**) The expression patterns of all expressed genes in the Wnt pathway in Japanese flounder during gametes and embryonic development. M: sperm, F: egg, 4c: 4-cell, 32c: 32-cell, 128c: 128-cell, HB: high blastula, LB: low blastula, EG: early gastrula, LG: late gastrula, NE: neurula, HBT: heart-beating, HS: hatching stage.

## Data Availability

The transcriptome data used in this study were uploaded to the China National GeneBank (CNGB) (https://db.cngb.org/, accessed on 29 September 2021) with accession numbers CNP0000304.

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
