# Peer review of "Temporal Transcriptome Analysis Reveals Dynamic Expression Profiles of Gametes and Embryonic Development in Japanese Flounder (Paralichthys olivaceus)"

_genes, 2021, doi:10.3390/genes12101561_

Round 1
Reviewer 1 Report
This is an interesting study on the transriptome in the maternal-to-zygotic transition. Japanese flounder is the species studied. There are some comparisons to other "flatfish". However, it should be noted that this is not one single phylogenetic group, and that if comparisons are to be made, they should be specific towards various phylogenetic groups.
My main concern is that the manuscript needs to have its English corrected.
The general setup and the literature work is sound.
Author Response
Dear Editor and Reviewers
We are grateful to have been given the opportunity to revise our manuscript, which we now entitle “Temporal transcriptome analysis reveals dynamic expression profiles of gametes and embryonic development in Japanese flounder (Paralichthys olivaceus)”. We want to extend our appreciation for the time and effort that you and the reviewers have dedicated to providing such valuable guidance on our manuscript. Based on these comments and suggestions, we have made careful modifications to the original manuscript.
Point1: My main concern is that the manuscript needs to have its English corrected.
Response 1: We apologize for the language problems in the original manuscript. The language presentation was improved with assistance from a native English speaker with an appropriate research background.
Point2: However, it should be noted that this is not one single phylogenetic group and that if comparisons are to be made, they should be specific towards various phylogenetic groups.
Response 2: We are grateful for the suggestion. In the previous studies, the genes involved in MZT are very conserved among different species, and zebrafish and Xenopus have unique representations in their phylogenetic group, so we selected these two species to compare with Japanese flounder to explore the time when MZT occurred.

Reviewer 2 Report
In the current manuscript, the authors investigated the transcriptomes of Japanese flounder gametes and embryonic stages to study the maternal-to-zygotic transition (MZT). Samples were collected from gametes and 10 embryonic stages at different time intervals. RNA-seq was performed and the data was analyzed to detect the differentially expressed genes. In general, the study provides important information on the MZT and key genes that are expressed at this stage in Japanese flounder. Below, are my comments and suggestions for the authors.
General comment for the entire manuscript:
This manuscript will definitely require language editing. In many sentences, it is very hard to understand the meaning due to language.
Abstract:
Line 18: the expression of which genes decreased? Maternal factors? Please clarify.
Line 21: The abstract should stand alone, so please define ZGA.
Line 25: The authors did not study other species, therefore the conclusion should include only what was done and is supported by the results. Also, the authors state here that the MZT is conserved among species but in line 42, they said that MZT is not conserved. Which one is correct?
Introduction:
Line 32: mitochondria is an organelle. Please use “mitochondrial genes” instead.
Line 33-34: completely different in structure and function from what? This sentence is vague.
Lines 70-72: These are results and should not be included here.
The rationale of the experiment need to be clearly justified.
Materials and Methods:
Line 75: What does the authors mean by “two gametes”? Only two gametes? Or two types of gametes, the sperm and egg? Please clarify.
Lines 81-82: Please provide the details of artificial spawning of male and female parent fish or provide a reference. What was the age of broodstock fish?
Lines 85-89: How many embryos were collected for each stage? Was the embryonic development examined before sample collection to ensure that the eggs are fertilized and have developing embryos? Please clarify.
Line 94: the total RNA was qualified?
Line 95: How was the library constructed? Please provide the details or a reference.
Lines 100-103: What is the filtering threshold? Other parameters for trimming? Please provide the details or cite a reference. Also, please provide a reference for SOAPnuke.
Lines 106-107: Please delete the repeated sentence. Also, add a reference for normalization.
Results and discussion:
Line 155: There is no information mentioned in the methods section about the use of Mfuzz. Please add this information to the methods section and include the software version used.
Lines 181-182: 4,127 is not nearly twice 2,746. Please check.
General comment for Figures:
The quality of figures is not good for publication. In some figures (such as Fig 2B, Fig 5, … ), it is very hard to read the figure because of the low quality. Please provide publication quality figures. Also, please define the abbreviations in the figure legends (example, DEGs in Figure 2 legend, … etc.).
Fig 1A is not cited in the text. Legend: “The diagram of the developmental stages of the embryo in Japanese flounder. (A): A scheme of Japanese flounder embryonic developmental stages”.. These two sentences are almost the same. I suggest deleting one of them.
The resolution of figure 5 is very bad. It is too difficult to read.
Line 241: Do maternal genes really degrade with embryonic development? Or the maternal gene products (RNA and other maternal factors)? This is confusing because what changes are the gene products not the genes themselves. This should be corrected here and in other parts of the manuscript.
Supplementary Table:
Please provide a title for the table. Also, the sample names should be defined and the typos (row) corrected.
Author Response
Dear Editor and Reviewers
Thank you for offering the insightful and valuable comments on our manuscript titled “Temporal transcriptome analysis reveals dynamic expression profiles of gametes and embryonic development in Japanese flounder (Paralichthys olivaceus)”. Those comments are valuable and very helpful. We have read through the comments carefully and have made corrections. Based on the instructions provided in your letter, we uploaded the file of the revised manuscript.
Based on the comments and suggestions, we have revised the manuscript and addressed all the issues raised, item-by-item, as detailed below.
Point 1: This manuscript will definitely require language editing. In many sentences, it is very hard to understand the meaning due to language.
Response 1: We apologize for the language of our manuscript. We worked on the manuscript for a long time and the repeated addition and deletion of sentences obviously led to poor readability. We have now carefully checked the full text and have involved a professional English language editing service for polishing.
Point 2: Line 18: the expression of which genes decreased? Maternal factors? Please clarify.
Response 2: We are grateful for the suggestion. To be more clear, we have added the maternal factor-related genes in line 18.
Point 3: Line 21: The abstract should stand alone, so please define ZGA.
Response 3: Thanks for the reviewer's suggestions. The ZGA was defined as zygotic genome activation was added in line 22.
Point 4: Line 25: The authors did not study other species, therefore the conclusion should include only what was done and is supported by the results. Also, the authors state here that the MZT is conserved among species but in line 42, they said that MZT is not conserved. Which one is correct?
Response 4: I think our ambiguous description may confuse the reviewer. What I want to describe is that the genes involved in MZT are conservative, but the occurrence time of MZT is not conservative. Also, in line 25 we deleted the research results about other species. According to the reviewer’s opinion, we modify the sentence to “In conclusion, the results report the list of the maternal gene and definition the stage of MZT in Japanese flounder, which will contribute to understanding the details of MZT during Japanese flounder embryo development” in line 24-26.
Point 5: Line 32: mitochondria is an organelle. Please use “mitochondrial genes” instead.
Response 5: Thanks for the reviewer’s remind. We agree with the comment and re-wrote the mitochondrial genes in line 33.
Point 6: Line 33-34: completely different in structure and function from what? This sentence is vague.
Response 6: We apologize for the ambiguous description that may confuse the reviewer. They completely difference in structure and function from fertilized egg to larvae. We have revised it in line 34-35.
Point 7: Lines 70-72: These are results and should not be included here. The rationale of the experiment needs to be clearly justified.
Response 7: Thanks for the reviewer’s reminding. We have made corrections according to the reviewer’s specific comment. Line 67-73 was changed into “Previous studies revealed extensive variability stages among different species of maternal gene product degradation and zygotic gene activation. However, the stage of MZT in Japanese flounder (Paralichthys olivaceus) remains unknown.”
Point 8: Materials and Methods: Line 75: What do the authors mean by “two gametes”? Only two gametes? Or two types of gametes, the sperm and egg? Please clarify.
Response 8: Thanks for the reviewer’s remind. That was mean “sperm and egg” and rewriting in line 82.
Point 9: Lines 81-82: Please provide the details of artificial spawning of male and female parent fish or provide a reference. What was the age of broodstock fish?
Response 9: Thanks for the reviewer’s suggestion. The detail of the artificial spawning of male and female fish are introduced as follows, “The sperm was collected using a 5-mL plastic syringe by gently pressing on the abdomen, thus stored in the dark at 4 °C until used. The eggs were collected by gently stripping the abdomen of female fish into a 1000 mL glass beaker and stored in the dark at room temperature.” Show in line 85-88. And we used 3 years old broodstock fish.
Point 10: Lines 85-89: How many embryos were collected for each stage? Was the embryonic development examined before sample collection to ensure that the eggs are fertilized and have developing embryos? Please clarify.
Response 10: About 1000 embryos were collected at each stage for each group. Before sampling, dead embryo were removed and developmental stages were observed under stereomicroscope (Olympus). That was added in line 94-96.
Point 11: Line 94: the total RNA was qualified?
Response 11: Thanks for reviewer’s suggestion, that are rewritten as “the total RNA was qualified and quantified” That was added in line 101.
Point 12: Line 95: How was the library constructed? Please provide the details or a reference.
Response 12: Thanks for reviewer’s suggestion. A total 46 cDNA libraries were constructed were used the NED kit according to the manufacturer’s instructions, the high-quality RNA was used 3 μg RNA (RIN > 7.0) from each sample for mRNA library construction by means of the conventional protocol. That was in line 102-106.
Point 13: Lines 100-103: What is the filtering threshold? Other parameters for trimming? Please provide the details or cite a reference. Also, please provide a reference for SOAPnuke.
Response 13: Thanks for the reviewer’s suggestion. We removed adapters and filter low-quality reads (the rate of reads whose quality value ≤ 10 is more than 20%) by SOAPnuke (version 1.4.0); the reads with unknown nucleotides ≥5% (options = -l 10 -q 0.1 -n 0.05 -Q 2 -G -1). That was in line 109-111. The reference is “Chen, Yuxin, Yongsheng Chen, Chunmei Shi, Zhibo Huang, Yong Zhang, Shengkang Li, Yan Li, Jia Ye, Chang Yu, and Zhuo Li. Soapnuke: A Mapreduce Acceleration-Supported Software for Integrated Quality Control and Preprocessing of High-Throughput Sequencing Data. Gigascience 2017, 7, 1: gix120”
Point 14: Lines 106-107: Please delete the repeated sentence. Also, add a reference for normalization.
Response14: We are grateful for the suggestion. We delete the repeated sentence and add the reference in line 117.
Point 15: Results and discussion: Line 155: There is no information mentioned in the methods section about the use of Mfuzz. Please add this information to the methods section and include the software version used.
Response 15: Thank you for your comment, we are using the R package of Mfuzz (version 2.52), and have added a more detailed in line 120-122.
Point 16: Lines 181-182: 4,127 is not nearly twice 2,746. Please check.
Response 16: We are grateful for the suggestion. The sentence has been modified to “4,127 were more 1.5-fold higher than 2,746” detail in line 202-203.
Point 17: General comment for Figures:
1)The quality of the figures is not good for publication. In some figures (such as Fig 2B, Fig 5, … ), it is very hard to read the figure because of the low quality. Please provide publication quality figures. Also, please define the abbreviations in the figure legends (example, DEGs in Figure 2 legend, … etc.).
2)The resolution of figure 5 is very bad. It is too difficult to read.
3)Fig 1A is not cited in the text. Legend: “The diagram of the developmental stages of the embryo in Japanese flounder. (A): A scheme of Japanese flounder embryonic developmental stages”. These two sentences are almost the same. I suggest deleting one of them.
Response 17: Thanks a lot for reviewer’s suggestion. We have made corrections according to the reviewer’s specific comment as following. The point 1 and point 2, we also have re-upload the high-resolution image; Also, we added detailed abbreviations in the figure legends. The point 3 (Fig 1) legends we re-write as “The samples were used in the present study.” in line 145-149.
Point 18: Line 241: Do maternal genes really degrade with embryonic development? Or the maternal gene products (RNA and other maternal factors)? This is confusing because what changes are the gene products not the genes themselves. This should be corrected here and in other parts of the manuscript.
Response 18: Thanks for your suggestions. We agree with the comment and re-wrote the “maternal gene products”. We also carefully checked the full text to make corrections. We hope our changes can make the manuscript more accurate and focused.
Point 19: Supplementary Table: Please provide a title for the table. Also, the sample names should be defined and the typos (row) corrected.
Response 19: Thanks for your comments. We provide a title as “Statistic of transcription data in all samples”; the names of the sample defined as 2P and 2M indicate different family; F indicates eggs; M indicates sperm; 4c: 4-cell, 32c: 32-cell, 128c: 128-cell, HB: high blastula, LB: low blastula, EG: early gastrula, LG: late gastrula, NE: neurula, HBT: heart-beating, HS: hatching stage; The “row” to correct for “raw”.
